# High-Performance Chlorine-Doped Cu₂O Catalysts for the Ethynylation of Formaldehyde

**Jie Gao [1,2], Guofeng Yang [3], Haitao Li [3,*] , Mei Dong [1], Zhipeng Wang [3] and Zhikai Li [1]**

[1]  State Key Laboratory of Coal Conversion, Institute of Coal Chemistry, Chinese Academy of Sciences, P.O. Box 165, Taiyuan 030001, China; jgao2006@163.com (J.G.); mdong@sxicc.ac.cn (M.D.); lizhikai@sxicc.ac.cn (Z.L.)

[2]  University of Chinese Academy of Sciences, Beijing 100049, China

[3]  Research Center for Fine Chemical Engineering of the Ministry of Education, School of Chemistry and Chemical Engineering, Shanxi University, Taiyuan 030006, China; guofengyang0351@163.com (G.Y.); zhipengw44@163.com (Z.W.)

\*  Correspondence: htli@sxu.edu.cn; Tel.: +86-136-0353-6381

**Abstract:** The in situ formed $Cu^+$ species serve as active sites in the ethynylation of formaldehyde. The key problem that needs to be solved in this process is how to avoid excessive reduction of $Cu^{2+}$ to inactive metallic Cu, which tends to decrease the catalytic activity. In this work, $Cl^-$-modified $Cu_2O$ catalysts with different Cl content were prepared by co-precipitation. The characterization results demonstrated that $Cl^-$ remained in the lattice structure of $Cu_2O$, inducing the expansion of the $Cu_2O$ lattice and the enhancement of the Cu–O bond strength. Consequently, the reduction of $Cu^+$ to $Cu^0$ was effectively prevented in reductive media. Moreover, the activity and stability of $Cu_2O$ were significantly improved. The $Cl^-$ modification increased the yield of 1,4-butynediol (BD) from 73% to 94% at a reaction temperature of 90 °C. More importantly, the BD yield of $Cl^-$ modified $Cu_2O$ was still as high as 86% during the ten-cycle experiment, whereas the BD yield of $Cu_2O$ in the absence of $Cl^-$ decreased sharply to 17% at the same reaction conditions. This work provides a simple strategy to stabilize $Cu^+$ in reductive media.

**Keywords:** $Cu_2O$; $Cl^-$ modification; reducibility; formaldehyde ethynylation; 1,4-butynediol

## 1. Introduction

Catalytic ethynylation of formaldehyde is an important initial process for mass production of high-value intermediates, such as 1,4-butynediol (BD), propynol (PO), and downstream chemicals, such as 1,4-butanediol (BDO), 3-butene-1-alcohol (BTO), tetrahydrofuran (THF), poly (tetramethylene ether) glycol (PTMEG), γ-butyrolactone (GBL), polybutylene succinate (PBS), and polybutylene terephthalate (PBT). These compounds are in high demand in various industries, such as pharmaceutical, textile, military, electrical/electronics, automotive manufacturing, and others [1–8].

The catalysts used in the ethynylation of formaldehyde are mainly Cu-based catalysts, including artificial malachite catalysts, CuO powder catalysts, and supported CuO catalysts [9–18]. Under the reaction conditions, these initial $Cu^{2+}$ compounds are converted in situ to active species (cuprous acetylide) and undergo some complex chemical changes: (1) $Cu^{2+}$ is selectively reduced to $Cu^+$ by formaldehyde; and (2) $Cu^+$ reacts with acetylene to produce cuprous acetylide [17–19]. The key problem that needs to be solved in this process is how to avoid the excessive reduction of $Cu^{2+}$ to metallic Cu by the reductive medium (formaldehyde/acetylene). Namely, metallic Cu will cause the formation of polyacetylene by-products and rapid deactivation of the catalyst [17–19]. Literature mainly reports introducing $Bi_2O_3$ to inhibit over-reduction of $Cu^{2+}$ [15,19,20].

In recent years, our research group has found that the over-reduction of $Cu^{2+}$ can be effectively inhibited by the electronic assistant effect of $Fe_3O_4$ and the strong interaction between Cu and $SiO_2$-MgO aerogel support. This is beneficial to the transformation of $Cu^{2+}$ to active $Cu^+$ species and their stability in the reaction atmosphere [21,22]. When $Cu_2O$ was directly applied to the ethynylation of formaldehyde, the production of metallic Cu was still unavoidable. Moreover, it was still necessary to stabilize $Cu^+$ in the reductive medium by controlling the grain size of $Cu_2O$ and regulating the interactions between $Cu_2O$ and the support [23–25]. However, even when the above measures were effective, the decrease in the catalytic activity caused by the reduction of $Cu^+$ to metallic Cu could not be avoided during prolonged operation.

The stability of $Cu^+$ has also attracted much attention in many reaction systems even using metal $Cu^0$ as catalyst [26–28]. The initial structure, chemical environment, and electronic configuration of $CuO/Cu_2O$, such as preparing copper silicate precursors, molecular sieves exceeding cage confinement, and controlling exposed crystal planes [28–32]. This further determines the stability and catalytic performance of $Cu^+$ species in a reductive atmosphere. Recent studies have shown that adding $Cl^-$ can also regulate the structure and morphology of $Cu_2O$. Wang et al. [33] reported a surfactant-free synthesis of hollow $Cu_2O$ nanocubes by reducing the $Cu^{2+}$ precursors with only $Cl^-$ ions as the morphology regulators. The $Cu_2Cl(OH)_3$ intermediate formed in the presence of $Cl^-$ played a decisive role in controlling the morphology of $Cu_2O$. Yang et al. [34] prepared bi-component $Cu_2O$–CuCl composites by introducing a Cl source into a simple hydrothermal reduction. They found that $Cu_2O$–CuCl composites showed more numerous oxygen vacancies and narrower band gaps compared with the single-component $Cu_2O$ sample, ultimately enhancing the photodecoloration performance of methylene blue. Changes in the $Cu_2O$ structure and morphology in the above studies inevitably affected the chemical environment of $Cu^+$ species. However, it has not yet been reported in the literature whether $Cl^-$ doping also affects the stability and catalytic performance of $Cu^+$ in the acetylation of formaldehyde.

In the present work, $Cl^-$-modified $Cu_2O$ microcrystals were prepared using copper nitrate as a copper source and NaCl as a Cl source. The effects of $Cl^-$ on the structure, reducibility, and catalytic performance of $Cu_2O$ in the ethynylation of formaldehyde were studied.

## 2. Materials and Methods

### 2.1. Preparation of $Cu_2O$ Catalysts

$Cl^-$-modified $Cu_2O$ samples were prepared with $Cu(NO_3)_2 \bullet 3H_2O$ (AR, Sinopharm Chemical Reagent Co., Ltd., Shanghai, China) as a Cu precursor and NaCl (AR, Sinopharm Chemical Reagent Co., Ltd., Shanghai, China) as a $Cl^-$ source. In a typical synthesis, 100 mL of copper nitrate solution (1.25 mol/L) was mixed with 100 mL of NaCl solution (1.25 mol/L) and 100 mL of polyethylene glycol 600 (average Mn 600, Shanghai Macklin Biochemical Co., Ltd., Shanghai, China). In addition, 150 mL of sodium hydroxide (AR, Tianjin Tianxin Fine Chemical Development Center, Tianjin, China) solution (3.33 mol/L) was added to the above solution under stirring at 30 °C, and a dark-blue colloid immediately formed. Five minutes later, 300 mL of sodium L-ascorbate (AR, TCI (Shanghai) Development Co., Ltd., Shanghai, China) solution (0.075 mol/L) was added dropwise into the colloidal suspension; the dropping rate was 1.25 mL/min. The resulting precipitate was separated by centrifugation and washed several times with deionized water and ethanol. Finally, the obtained products were dried in a vacuum at 60 °C for 4 h and then calcined at 300 °C for 3 h in an $N_2$ atmosphere. The as-synthesized $Cu_2O$ was denoted as $Cu_2O$-Cl(2). The $Cu_2O$ with lower $Cl^-$ content was prepared by decreasing the NaCl solution concentration to 0.6 mol/L (while all other conditions remained unchanged). The obtained product was denoted as $Cu_2O$-Cl(1). As a contrast, $Cu_2O$-$NO_3$ was prepared under the same reaction conditions as $Cu_2O$-Cl(1) and $Cu_2O$-Cl(2) but without adding the NaCl solution. As shown in Table 1, the $Cl^-$ content in $Cu_2O$-Cl(1) and $Cu_2O$-Cl(2) was 1.1% and 3.1%, respectively.

**Table 1.** Lattice spacing (d), crystallite size (D) and Cl content of the samples.

| Samples | $2\theta$ (°) | FWHM (°) | d-Spacing [a] (Å) | D [b] (nm) | Cl Content [c] (%) |
|---|---|---|---|---|---|
| $Cu_2O-NO_3$ | 36.4633 | 0.141 | 2.4612 | 58.7 | – |
| $Cu_2O-Cl(1)$ | 36.4487 | 0.148 | 2.4621 | 55.9 | 1.1 |
| $Cu_2O-Cl(2)$ | 36.4322 | 0.158 | 2.4632 | 52.3 | 3.1 |

[a] Calculated according to the formula: $n\lambda = 2d\sin\theta$, [b] Calculated according to the formula: $D = K\lambda/\beta\cos\theta$, [c] Measured by X-ray fluorescence spectroscopy shXRF).

## 2.2. Characterization of $Cu_2O$ Catalysts

X-ray diffraction (XRD) patterns of the samples were recorded with a D8 Advance diffractometer (Bruker, Billerica, MA, USA) with Cu K$\alpha$ radiation ($\lambda$ = 1.5418 Å). X-ray fluorescence spectroscopy (XRF) measurements were performed on an S8 Tiger spectrometer (Bruker, Billerica, MA, USA). Fourier-transform infrared (FTIR) spectra were recorded with a Nicolet™ iS50 spectrophotometer (Thermo Fisher, Waltham, MA, USA) in the range of 400–4000 cm$^{-1}$. Raman spectroscopy was performed with a LabRAM HR Evolution Raman spectrograph (HORIBA, Tokyo, Japan) with a 532 nm laser operated at 0.08 mW. Scanning electron microscopy (SEM) images were obtained with an S-4800 electron microscope (Hitachi, Tokyo, Japan) operated at a beam energy of 3.0 kV. X-ray photoelectron spectroscopic (XPS) measurements were conducted on an ESCALAB 250 spectrometer (Thermo Fisher, Billerica, MA, USA) using an Al K$\alpha$ X-ray source (h$\nu$ = 1486.7 eV). All spectra were referenced to adventitious C (1s) at a binding energy of 284.5 eV. X-ray excited Auger electron spectroscopy (XAES) was performed on a PHI 1600 ESCA spectrometer (Perkin-Elmer, Waltham, MA, USA) equipped with a monochromatic Al Ka X-ray source (hm = 1361 eV) operating at a pass energy of 100 eV. $H_2$ temperature-programmed reduction ($H_2$-TPR) experiments were performed on an AutoChem II 2920 automatic temperature-programmed chemical adsorption instrument (Micromeritics, Norcross, GA, USA). Approximately 30 mg of the sample was loaded into a quartz U-tube for each measurement. Prior to the measurement, the sample was heated to 300 °C for 30 min under an Ar stream. When the temperature was dropped to 50 °C, the $H_2$–Ar mixture (5% $H_2$ by volume) was switched on, and the temperature was increased to 400 °C with a ramp of 10 °C/min. Hydrogen consumption was measured by a Thermal Conductivity Cell Detector. The Cu K-edge X-ray absorption fine structure (XAFS) measurements were performed at the Beijing Synchrotron Radiation Facility (BSRF), using a Si (111) double-crystal monochromator. The storage ring was operated at 3.5 GeV with a maximum current of 300 mA. The absolute energy position was calibrated using a Cu metal foil. XAFS data processing and analysis were performed using the IFEFFIT package [35]. EXAFS analysis was done model-independently, and the results were not biased in favor of any assumed model for the short-range order of elements in these samples. Specifically, multiple-edge (Cu-K) analysis was employed by fitting theoretical FEFF6 calculations to the experimental EXAFS data in R-space. The values of passive electron reduction factor, $S_0^2$, were obtained to be 0.80 by fitting to their corresponding standards and were fixed in the analysis of the Cu samples. The data used in the EXAFS fit ranged from k = 3.5 to 14 Å$^{-1}$. The fitting was done in R-space in the range of 1.0–3.5 Å with a sine window and multiple kn weighting ($n$ = 2). The parameters describing electronic properties (e.g., correction to the photoelectron energy origin) and local structural environment (coordination numbers, N, bond length, R, and their mean-squared relative derivation, $\sigma^2$) around the absorbing atoms were varied during fitting.

## 2.3. Testing of $Cu_2O$ Catalysts

The evaluation of the catalysts was carried out in a three-neck flask connected to a reflux condenser. In addition, 0.5 g of the catalyst and 50 mL of and aqueous formaldehyde solution (39 wt %, Zhangjiakou Chemical Reagent Factory, Zhangjiakou, China) were mixed in the flask and placed in an oil bath with electromagnetic stirring. A flow of pure $N_2$ (purity > 99.99%, Taiyuan Taineng Gas Co., Ltd., Taiyuan, China) was introduced into the flask to purge $O_2$, and then the catalyst-containing solution was heated to reaction temperature (90 °C) under continuous stirring. Subsequently, a $C_2H_2$

stream (purity > 99.99%, Taiyuan Taineng Gas Co., Ltd., Taiyuan, China) was switched on to start ethynylation under atmospheric pressure. 24 h later, the catalytic reaction was stopped by decreasing the reaction temperature to room temperature, introducing the $N_2$ stream, and closing the $C_2H_2$ stream. The used catalyst was centrifuged, washed with deionized water, and dried in a vacuum. The formaldehyde content ($x_{HCHO}$) in the centrifugate was determined by titration with sodium thiosulfate to obtain the formaldehyde conversion. The BD content ($x_{BD}$) was analyzed on an Agilent 7890A gas chromatograph (Agilent Technologies, Santa Clara, CA, USA) using a 1,4-butanediol-added internal standard. Moreover, the yield of BD was calculated as $Y_{BD} = (m_1 \times x_{BD} \times 2 \times M_{HCHO})/(m_0 \times x_{HCHO} \times M_{BD}) \times 100\%$, where $m_1$ was the total mass (g) of the BD solution after reaction, $m_0$ was the total mass (g) of the initial formaldehyde solution, "2" was the ratio of formaldehyde to BD in the reaction formula ($HC{\equiv}CH + 2HCHO \rightarrow HO\text{-}CH_2C{\equiv}CCH_2\text{-}OH$), $M_{HCHO}$ was the molecular weight of formaldehyde, and $M_{BD}$ was the molecular weight of BD.

## 3. Results

### 3.1. Characterization of Cu$_2$O Catalysts

#### 3.1.1. The Structure of Cu$_2$O Catalysts

The XRD patterns of the as-synthesized $Cu_2O$ microcrystals are shown in Figure 1. As can be seen, all samples show similar diffraction patterns. The diffraction peaks with $2\theta$ values of 29.6°, 36.5°, 42.4°, 61.4°, 73.6°, and 77.4° correspond to (110), (111), (200), (220), (311) and (222) planes of $Cu_2O$, respectively, which can be exactly indexed to the standard cubic phase $Cu_2O$ (JCPDS No. 05-0667). No other crystalline impurities of metallic Cu, CuO, or other species were detected in these patterns. The mean crystallite sizes along the (111) plane of different $Cu_2O$ catalysts are summarized in Table 1. As can be seen, the $Cl^-$-modified $Cu_2O$ showed smaller mean crystallite size compared with $Cu_2O$-NO$_3$, indicating poor crystallinity of the $Cl^-$-modified $Cu_2O$.

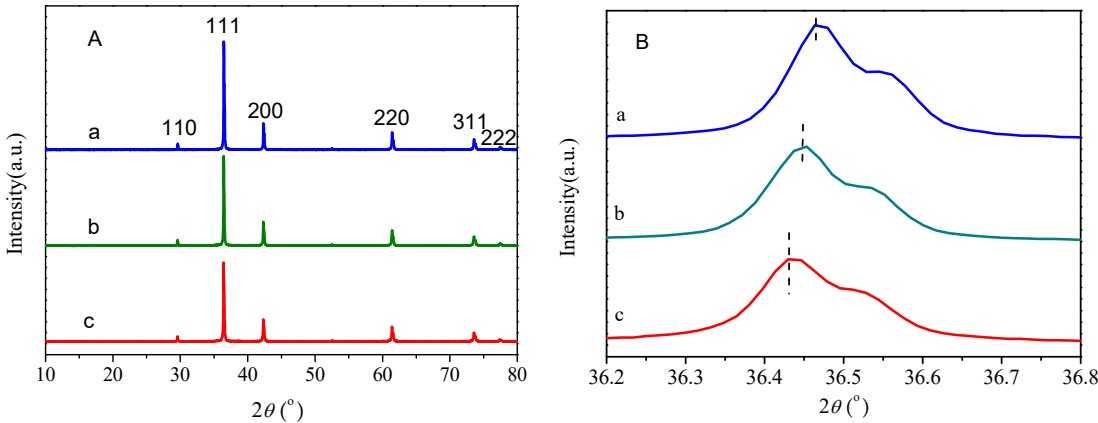

**Figure 1.** X-ray diffraction (XRD) patterns of $Cu_2O$-NO$_3$ (**a**), $Cu_2O$-Cl(1); (**b**) and $Cu_2O$-Cl(2) (**c**). ((**A**): wide angle XRD patterns; (**B**): magnified $Cu_2O$ (111) diffraction peak).

By amplifying the (111) diffraction peak of $Cu_2O$ (Figure 1B), the peak positions of $Cu_2O$-Cl(1) and $Cu_2O$-Cl(2) shift slightly to lower angle values. We calculated the interplanar spacing, 'd,' (d-spacing), which is also summarized in Table 1. The d-spacing for (111) plane of the $Cl^-$-modified $Cu_2O$ was higher than that of $Cu_2O$-NO$_3$ because the changes in the bond length and angle between the Cu and O atoms generated lattice strain. It is speculated that $Cl^-$ replaces part of the O atoms in the $Cu_2O$ lattice, causing the lattice to expand as the atomic radius of $Cl^-$ is larger than that of O [36].

The structure of $Cu_2O$ was also characterized by FTIR and Raman spectroscopy. FTIR spectra of the three $Cu_2O$ samples (Figure 2) exhibit a characteristic band at 631 cm$^{-1}$, which belongs to the vibrational mode of Cu–O in the $Cu_2O$ phase [37]. The corresponding Raman spectra (Figure 3)

show characteristic peaks of $Cu_2O$ at wavenumbers of 94 cm$^{-1}$, 148 cm$^{-1}$, 218 cm$^{-1}$, 414 cm$^{-1}$, and 631 cm$^{-1}$ [38]. These results indicate that the preparation of $Cu_2O$ with copper nitrate as a Cu source and NaCl as a Cl$^-$ source was successful.

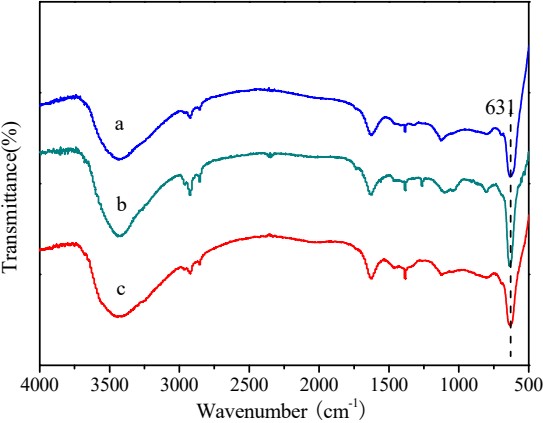

**Figure 2.** Fourier-transform infrared (FTIR) spectra of (**a**) $Cu_2O$-$NO_3$, (**b**) $Cu_2O$-Cl(1), and (**c**) $Cu_2O$-Cl(2).

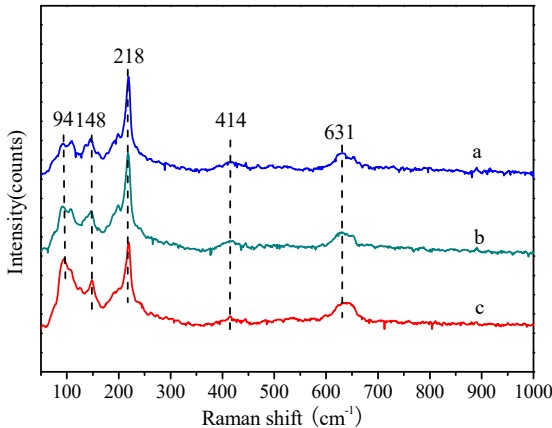

**Figure 3.** Raman spectra of (**a**) $Cu_2O$-$NO_3$, (**b**) $Cu_2O$-Cl(1), and (**c**) $Cu_2O$-Cl(2).

### 3.1.2. The Morphology of $Cu_2O$ Catalysts

Modification with Cl$^-$ may play a significant role in controlling the morphology and particle size of $Cu_2O$ microcrystals. The morphology and particle size of $Cu_2O$ microcrystals was characterized by SEM. As shown in Figure 4a,b, $Cu_2O$-$NO_3$ showed a 26-facet polyhedral octahedron shape with three pairs of {100} facets, four pairs of {111} facets, and six pairs of {110} facets. All facets had smooth surfaces, indicating a complete crystallization of $Cu_2O$. Four pairs of {111} facets with a smooth surface were observed in $Cu_2O$-Cl(1), whereas the octahedra's edges and corners were full of small bumps, indicating that they were covered with many little particles (Figure 4c,d). All surfaces of the $Cu_2O$-Cl(2) octahedra were rough and covered with little particles (Figure 4e,f). The particle size of $Cu_2O$-$NO_3$ was similar to that of $Cu_2O$-Cl(1), namely around 1.5 μm, whereas the particle size of $Cu_2O$-Cl(2) was about 1 μm. The results show that the presence of Cl$^-$ inhibits the growth of the crystal plane and decreases the crystallinity of $Cu_2O$, which is consistent with the XRD results.

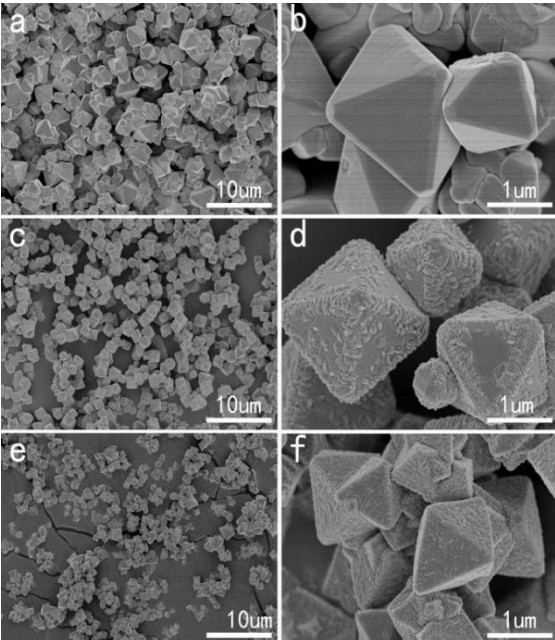

**Figure 4.** Scanning electron microscopy (SEM) images of (**a,b**) $Cu_2O$-$NO_3$, (**c,d**) $Cu_2O$-$Cl(1)$, and (**e,f**) $Cu_2O$-$Cl$ (2).

### 3.1.3. The Chemical Composition of the $Cu_2O$ Catalyst's Surface

Because of its extremely high sensitivity, XPS can acquire more detailed information on the chemical composition of the sample surface, especially on the corresponding anion residues. Figure 5A shows typical XPS spectra of the $Cu_2O$ samples. All of them contain Cu, O, and C. No N was observed near the binding energy of 400 eV in any of the three $Cu_2O$ samples. However, the Cl 2p peak near the binding energy of 200 eV was observed in $Cu_2O$-$Cl(1)$ and $Cu_2O$-$Cl(2)$. Moreover, the molar ratio of Cl to O was approximately 0.05 for $Cu_2O$-$Cl(1)$ and 0.12 for $Cu_2O$-$Cl(2)$ (Table 2). The results indicate that $NO_3^-$ was washed out during preparation or decomposed into corresponding oxides in the subsequent calcination and released in the form of a gas. However, $Cl^-$ remained in the lattice structure of $Cu_2O$, causing the lattice to expand, as shown by the XRD results.

High-resolution XPS spectra for Cu 2p and O 1s regions were used to evaluate in detail the surface electric state of the elements in the $Cu_2O$ samples. As shown in Figure 5B, the peaks positioned at a binding energy of 932.5 eV corresponded to Cu $2p_{3/2}$ and those at 952.4 eV to Cu $2p_{1/2}$ for $Cu_2O$ [39]. The peaks located at 933.7 eV were ascribed to Cu $2p_{3/2}$ and those at 953.7 eV to Cu $2p_{1/2}$ for CuO [40]. This confirms the coexistence of a trace amount of CuO [41], although it was not detected in the XRD, Raman, or FTIR measurements.

The $Cu^{2+}/Cu^+$ ratio was calculated from the peak area of Cu $2p_{3/2}$ for CuO and $Cu_2O$, and the results are shown in Table 2. It can be seen that the ratio values follow the sequence of $Cu_2O$-$Cl(2)$ > $Cu_2O$-$Cl(1)$ > $Cu_2O$-$NO_3$. The introduction of $Cl^-$ significantly increased the surface CuO species, which might be derived from the unreduced $Cu^{2+}(OH)_x$ [42]. This might affect the crystallization of $Cu_2O$ and result in its surface roughness (Figure 4f). Correspondingly, the surfaces of $Cu_2O$-$NO_3$, which had the lowest $Cu^{2+}/Cu^+$ ratio, were smooth (Figure 4b).

Figure 5C shows three different types of oxygen species in each sample. The peak observed at a binding energy of 529.8 eV is attributed to the lattice oxygen of CuO (denoted as $O\alpha$), and the one at 530.5 eV is attributed the lattice oxygen of $Cu_2O$ (denoted as $O\beta$). The peaks at a higher binding energy of 531.9 eV are assigned to the chemisorbed oxygen (denoted as $O\gamma$) in the surface oxygen vacancies [43–45]. The $O\gamma/(O\alpha + O\beta + O\gamma)$ ratio was calculated from the peak area of each oxygen species, and the data are shown in Table 2. As can be seen, $Cu_2O$-$Cl(2)$ had a much lower $O\gamma/(O\alpha + O\beta + O\gamma)$ ratio (0.58) than $Cu_2O$-$Cl(1)$ (0.68) or $Cu_2O$-$NO_3$ (0.72). Chemisorbed oxygen species are

more active than the lattice oxygen, which is attributed to the higher mobility of adsorbed oxygen [46]. It is speculated that the surface $O\gamma$ content will affect the reducibility of $Cu_2O$ [47].

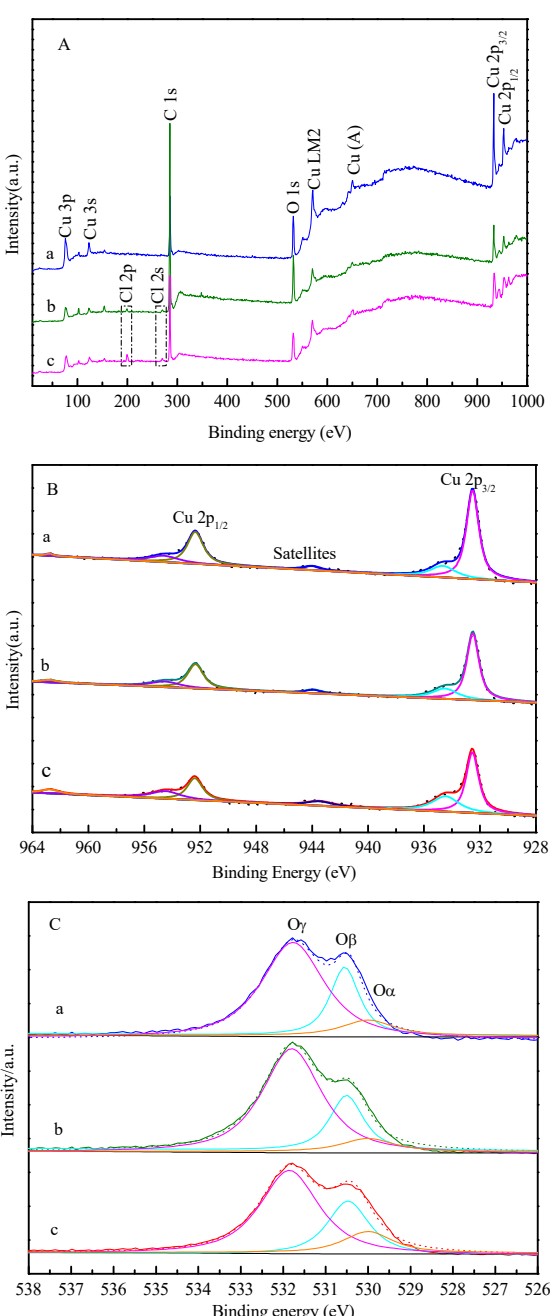

**Figure 5.** X-ray photoelectron spectroscopy (XPS) spectra of the three samples: (**A**) full survey spectrum, (**B**) Cu2p, and (**C**) O1s. (**a**) $Cu_2O-NO_3$; (**b**) $Cu_2O-Cl(1)$; and (**c**) $Cu_2O-Cl(2)$.

**Table 2.** The X-ray photoelectron spectroscopy (XPS) [a] analysis of surface chemical composition of the three samples.

| Samples | $Cu^{2+}/Cu^+$ | $O_\gamma/(O_\alpha + O_\beta + O_\gamma)$ | Cl/O |
|---|---|---|---|
| $Cu_2O-NO_3$ | 0.15 | 0.72 | – |
| $Cu_2O-Cl(1)$ | 0.18 | 0.68 | 0.05 |
| $Cu_2O-Cl(2)$ | 0.29 | 0.58 | 0.12 |

[a] Data based on a quantitative XPS analysis.

3.1.4. X-Ray Absorption Fine Structure (XAFS) Analysis of Cu$_2$O Catalysts

As shown by X-ray absorption near-edge spectroscopy (XANES) in Figure 6A, the samples have similar features to those of Cu$_2$O, whereas the peak shapes and positions for Cu$_2$O-Cl(2) appear slightly different from those of Cu$_2$O-NO$_3$. The Cu K-edge energy of Cu$_2$O-Cl(2) is shifted to 8981.1 eV from 8980.8 eV of Cu$_2$O-NO$_3$, indicating an increased Cu oxidation state. Moreover, the sharper shoulder peak (8982.7 eV) of the 1s to 4p electronic transition in Cu$_2$O is drastically decreased in Cu$_2$O-Cl(2) (Figure 6B). This indicates that a small amount of Cu$^+$ was lightly oxidized to Cu$^{2+}$, possibly by Cl$^-$ ions contained during the preparation [48], which is consistent with the XPS results.

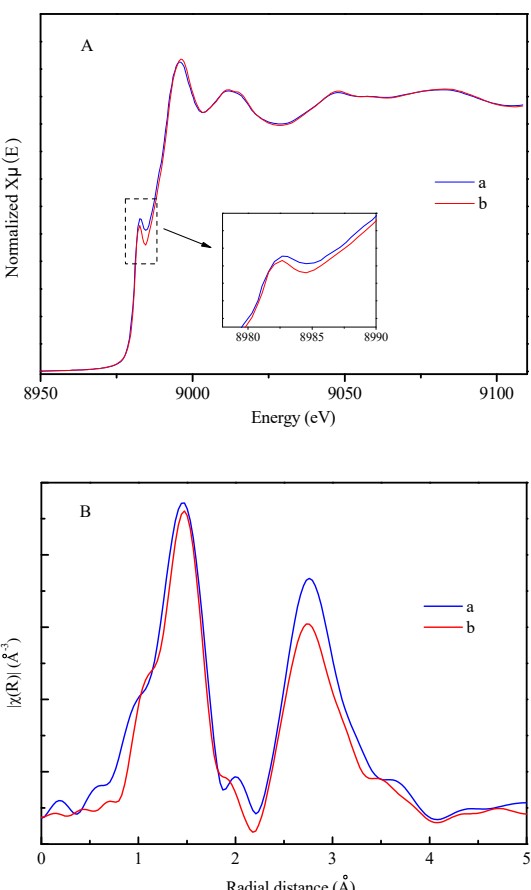

**Figure 6.** Cu K-edge data of Cu$_2$O samples. (**A**) X-ray absorption near-edge spectroscopy (XANES) and (**B**) extended X-ray absorption fine structure (EXAFS) Fourier transforms. (**a**) Cu$_2$O-NO$_3$; (**b**) Cu$_2$O-Cl(2).

To understand structural differences among the three catalysts, quantitative local structural properties were determined by fitting the extended X-ray absorption fine structure (EXAFS) data to the theoretical EXAFS calculations [49]. The EXAFS data were fitted using the IFEFFIT code [35,50] shown in Table 3. Local structural properties of Cu atoms in the three samples are similar to those of the Cu$_2$O bulk counterpart. However, the Cu–O bonds in Cu$_2$O-Cl(2) are slightly elongated. The change in bonding configuration can be ascribed to the additional bonding with the Cl$^-$ atom, which is in good agreement with the XRD results.

**Table 3.** Cu K-edge energy and extended X-ray absorption fine structure (EXAFS) fit parameters of the Cu$_2$O samples [a].

| Samples | Edge Energy (eV) | Contribution | N | R(Å) | $\Delta\sigma^2$(Å$^2$) ×10$^{-3}$ |
|---------|------------------|--------------|---|------|--------------------------------------|
| Cu$_2$O-NO$_3$ | 8980.79 | O | 2.0(1) | 1.82(7) | 2.(9) |
|                |         | Cu | 5.8(9) | 3.07(5) | 19.(1) |
| Cu$_2$O-Cl(2) | 8981.11 | O | 1.8(1) | 1.83(9) | 1.(8) |
|               |         | Cu | 5.7(4) | 3.08(0) | 19.(3) |

[a] N, coordination number; R, the distance between the absorber and backscattered atoms; $\Delta\sigma^2$, Debye–Waller factor. Error bounds (accuracies) characterizing the structural parameters obtained by EXAFS spectroscopy are estimated to be as follows: N, ~20%; R, ~0.02; $\Delta\sigma^2$, ~20%.

### 3.1.5. The Reducibility of Cu$_2$O Catalysts

Reduction properties of the three Cu$_2$O samples were determined by H$_2$-TPR, and the results are shown in Figure 7. Since there were no other reducible metal oxide components in the three Cu$_2$O samples, the hydrogen consumption peaks of each sample corresponded to the reduction of Cu$^+$ or a small amount of Cu$^{2+}$ to metallic Cu [51–53]. The reduction peaks reached the maximum H$_2$ consumption at 190 °C for Cu$_2$O-NO$_3$, 275 °C for Cu$_2$O-Cl(1) and 340 °C for Cu$_2$O-Cl(2). These results demonstrate that Cl$^-$-modified Cu$_2$O was much more difficult to reduce than Cu$_2$O-NO$_3$.

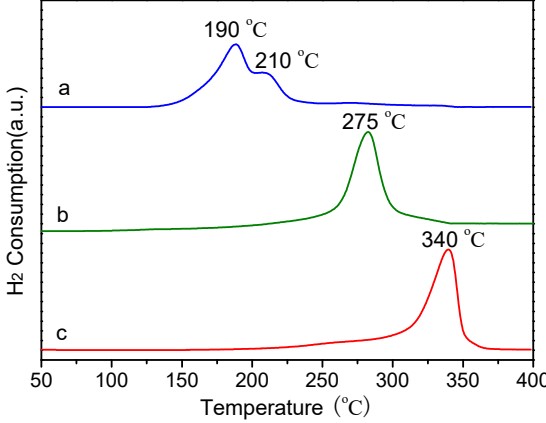

**Figure 7.** H$_2$ temperature-programmed reduction (H$_2$-TPR) profiles of (**a**) Cu$_2$O-NO$_3$, (**b**) Cu$_2$O-Cl(1) and (**c**) Cu$_2$O-Cl(2).

Generally, the reduction of oxides initially takes place on the surface or near-surface layer. The reduction of Cu$_2$O is affected by the surface composition and exposed crystal surface. Combined with the SEM and XPS characterization, Cl$^-$-modified Cu$_2$O showed a higher amount of surface CuO and different morphology compared with Cu$_2$O-NO$_3$. The apparent activation energy for the reduction of CuO is approximately 14.5 kcal/mol, whereas the value is 27.4 kcal/mol for Cu$_2$O. The reduction of CuO is easier than the reduction of Cu$_2$O [54]. Under a normal supply of H$_2$, CuO is reduced directly to metallic Cu, without a formation of an intermediate or suboxide (i.e., Cu$_4$O$_3$ or Cu$_2$O). High concentration of CuO on the surface of Cl$^-$-modified Cu$_2$O could not explain its high reduction temperature. Meanwhile, the small grain size of Cu$_2$O is beneficial to the reduction of Cu$_2$O [24], which is also contrary to the high reduction temperature of Cu$_2$O-Cl in the present work.

Many studies have shown that the reducibility of Cu$_2$O crystals was affected by surface oxygen vacancies and Cu–O bond strength [55]. It is reported that chemisorbed oxygen species are more active than the lattice oxygen [43,56]. As shown in the XPS results, Cl$^-$-modified Cu$_2$O samples had a lower concentration of adsorbed oxygen, which was one of the reasons for their high reduction temperature. In addition, previous reports have shown that Cl prefers to remain in Cu$_2$O [44,45]. Because of its strong electronegativity, Cl located in Cu$_2$O may generate the through-space electronic effect and

therefore strengthen the Cu–O bond strength. Copper tends to lose its electrons and is difficult to reduce [46]. A similar tendency to lose electrons in contact with the added Cl is also found in noble metal catalysts [48–50,54].

3.1.6. Structure and Surface Analysis of Used $Cu_2O$ Catalysts

In the ethynylation of formaldehyde, the initial $Cu^+$ species can be converted into cuprous acetylide in situ or transitively reduced to metallic Cu [14,16,17]. Thus, XRD, Raman, and XAES methods were used to characterize the changes in the structure and surface properties of used catalysts after a 24 h reaction. Before characterization, the catalysts were washed with deionized water and dried in a vacuum for 12 h at 30 °C.

The XRD patterns of used $Cu_2O$ samples are shown in Figure 8. Three diffraction peaks were detected at 2θ = 43.3°, 50.4°, and 74.1° in both samples, which corresponded to the fcc Cu (JCPDS04-0836) indicating that $Cu^+$ ions were partially reduced to metallic Cu during ethynylation. Compared with $Cu_2O$-$NO_3$, $Cu_2O$-Cl(2) exhibited weaker diffraction peaks of metallic Cu, indicating that $Cl^-$ can effectively inhibit the excessive reduction of $Cu^+$ to metallic Cu. According to the $H_2$-TPR characterization, it was attributed to the poor reducibility of $Cl^-$-modified $Cu_2O$.

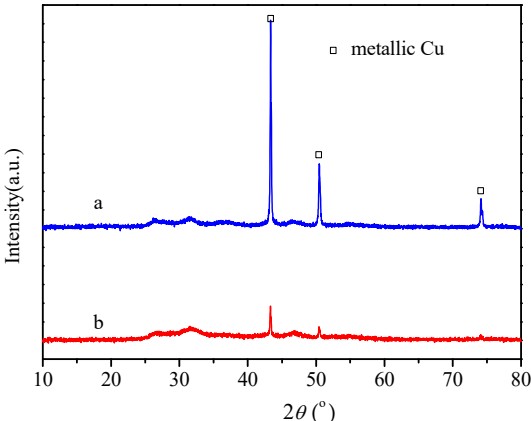

**Figure 8.** XRD patterns of used $Cu_2O$ samples after 24 h reaction. (**a**) $Cu_2O$-$NO_3$; (**b**) $Cu_2O$-Cl(2). Before characterization, the catalysts were washed with deionized water and then dried in a vacuum for 12 h at 30 °C.

We used Cu LMM XAES to investigate the distribution of $Cu^0$ and $Cu^+$. As shown in Figure 9, both $Cu_2O$-$NO_3$ and $Cu_2O$-Cl(2) showed two overlapping Cu LMM Auger kinetic energy peaks centered at approximately 918.4 eV ($Cu^0$) and 915.3 eV ($Cu^+$). Thus, $Cu^0$ and $Cu^+$ coexisted on the surface of the used catalysts. The ratio between $Cu^+$ and ($Cu^+ + Cu^0$) was obtained by the deconvolution of Cu LMM XAES spectra. As shown in Figure 9, the $Cu^+/(Cu^+ + Cu^0)$ ratio on the surface of $Cu_2O$-$NO_3$ was 0.63, which was lower than 0.94 of $Cu_2O$-Cl(2). This result further indicates that the $Cu^+$ in $Cu_2O$-Cl(2) was persistent, whereas the $Cu^+$ in $Cu_2O$-$NO_3$ was more easily reduced to metallic Cu under the reductive reaction conditions.

Raman spectra of used $Cu_2O$ samples are shown in Figure 10. Four significant Raman peaks were observed for $Cu_2O$-$NO_3$ at 1001 $cm^{-1}$, 1118 $cm^{-1}$, 1291 $cm^{-1}$, and 1595 $cm^{-1}$, which are attributed to the peaks of polyacetylene, indicating that polyacetylene was formed on the surface of $Cu_2O$-$NO_3$ during the reaction. However, only slight Raman peaks were observed at 1118 $cm^{-1}$ and 1595 $cm^{-1}$ in $Cu_2O$-Cl(2) indicating the existence of a very small amount of polyacetylene. In general, the formation of polyacetylene is due to two factors: (1) metallic Cu catalyzes the polymerization of acetylene and (2) the acidic center favors the formation of polyacetylene [20]. According to the XRD and XAES results of used samples, metallic Cu was observed in $Cu_2O$-$NO_3$, but only a small amount of metallic Cu existed

in Cu$_2$O-Cl(2). This metallic Cu may be responsible for promoting the formation of polyacetylene, which covered the surface of the catalyst, resulting in its deactivation.

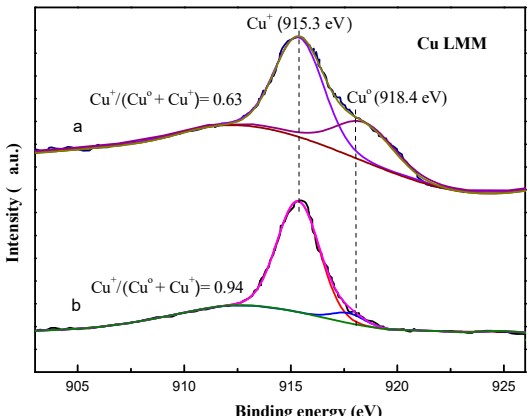

**Figure 9.** Cu LMM X-ray excited Auger electron spectroscopy (XAES) spectra of used Cu$_2$O samples after 24 h reaction. (**a**) Cu$_2$O-NO$_3$; (**b**) Cu$_2$O-Cl(2). Before characterization, the catalysts were washed with deionized water and dried in a vacuum for 12 h at 30 °C.

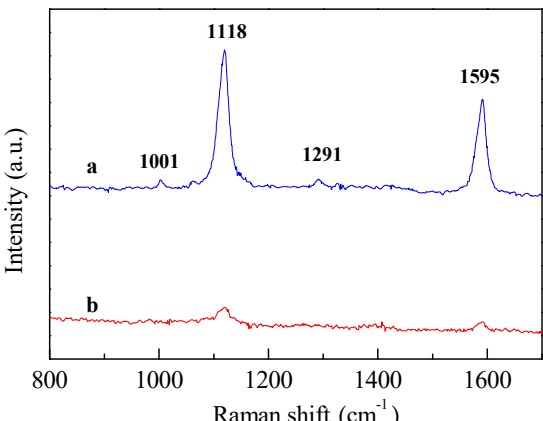

**Figure 10.** Raman spectra of used Cu$_2$O samples after a 24-h reaction. (**a**) Cu$_2$O-NO$_3$; (**b**) Cu$_2$O-Cl(2). Before characterization, the catalysts were washed with deionized water and then dried in a vacuum for 12 h at 30 °C.

### 3.2. Catalytic Performance

The reaction process of formaldehyde ethynylation can be described as follows:

$$HC\equiv CH + HCHO \rightarrow HO\text{-}CH_2C\equiv CCH_2\text{-}OH.$$

In the present work, only BD was observed in the gas chromatography analysis of the reaction products, and no other by-products such as propynol were observed. Therefore, the yield of BD was used to reflect the activity of the catalyst. The variation of BD yield with reaction time is shown in Figure 11. It can be seen that almost no BD was detected for any of the Cu$_2$O catalysts during the two-hour induction period. The presence of an induction period may be correlated with the phase evolution of the catalysts. After the induction period, the yield of BD increased with time for all Cu$_2$O catalysts. At 24 h, the yield of BD by Cu$_2$O-Cl(2) reached a maximum of 94%. Cu$_2$O-Cl(1) showed lower activity than Cu$_2$O-Cl(2) reaching a BD yield of approximately 81% at 24 h. On the other hand, the BD yield of Cu$_2$O-NO$_3$ was only about 73% at 24 h, which was significantly lower than that of the Cl$^-$-modified Cu$_2$O.

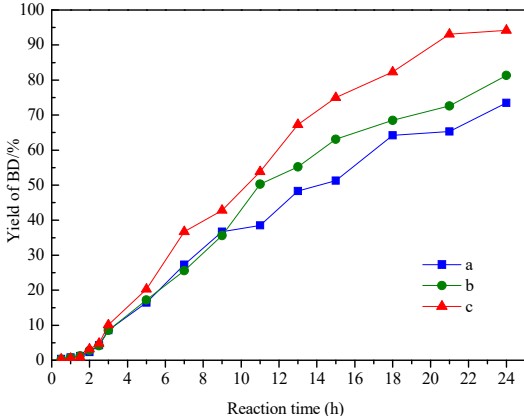

**Figure 11.** The yield of 1,4-butynediol (BD) by $Cu_2O$ catalysts. (**a**) $Cu_2O$-$NO_3$; (**b**) $Cu_2O$-Cl(1); (**c**) $Cu_2O$-Cl(2). Reaction conditions: catalyst amount, 0.5 g, formaldehyde solution concentration, 39 vol. %, consumption of the formaldehyde solution, 50 mL, reaction temperature, 90 °C.

The results of the stability evaluation of the $Cl^-$-modified $Cu_2O$ catalysts are shown in Figure 12. During ten cycles, a marginal decrease in BD yield was observed for $Cu_2O$ without $Cl^-$. The yield of BD decreased sharply from 73% to 17% after the ten-cycle experiment. Compared to $Cu_2O$ without $Cl^-$, the $Cl^-$-modified $Cu_2O$ showed higher stability. The BD yield of $Cu_2O$-Cl(1) decreased from 81% to 59%, whereas the BD yield of $Cu_2O$-Cl(2) decreased only from 94% to 86%. This indicates that $Cl^-$ modification significantly improved the stability of $Cu_2O$.

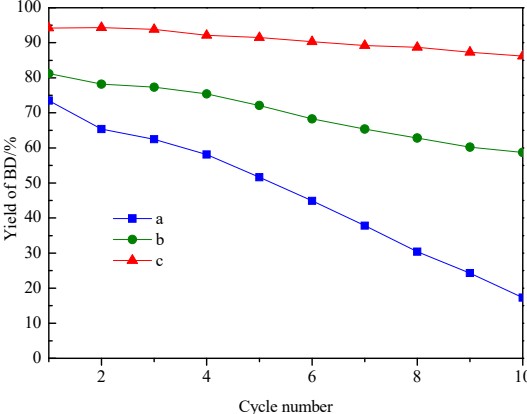

**Figure 12.** The yield of BD by $Cu_2O$ catalysts as a function of the cycle number. (**a**) $Cu_2O$-$NO_3$; (**b**) $Cu_2O$-Cl(1); (**c**) $Cu_2O$-Cl(2). Reaction conditions: catalyst amount, 0.5 g, formaldehyde solution concentration, 39 vol.%, consumption of the formaldehyde solution, 50 mL, reaction temperature, 90 °C, reaction time, 24 h.

## 4. Discussion

In the ethynylation of formaldehyde, the catalytic activity is proportional to the $Cu^+$ amount [20]. Based on the combination of characterization results, we speculate that the fundamental reason for different catalytic activity and stability is the difference in the $Cu_2O$ reduction performance caused by the $Cl^-$ modification. For $Cu_2O$-$NO_3$, $Cu^+$ ions were converted into active cuprous acetylide in situ. Meanwhile, the high reducibility of $Cu_2O$ (Figure 7) induced the production of a large amount of metallic Cu (Figures 8 and 9). This process directly led to the consumption of $Cu^+$ active centers. At the same time, polyacetylene catalyzed by metallic Cu covered the $Cu^+$ active centers, which further reduced the active $Cu^+$ exposure (Figures 8 and 9). Therefore, the high reducibility of $Cu_2O$ was the fundamental reason for the low activity and stability of $Cu_2O$-$NO_3$ catalysts. In $Cu_2O$-Cl, $Cl^-$ easily remains in the lattice structure of $Cu_2O$, decreasing the concentration of adsorbed oxygen and

strengthening the Cu–O bond. As a result, $Cu_2O$ was difficult to reduce, and $Cu^+$ persisted even in the reductive reaction conditions. The $Cl^-$-modified $Cu_2O$ catalysts thus showed higher catalytic activity and good stability.

Based on the comparison of the above analysis with the literature data, we believe that, when CuO is used as a catalyst precursor, its phase transformation during the acetylenization of formaldehyde includes (1) a direct reduction of CuO to metallic Cu by formaldehyde, and (2) the reduction of CuO to $Cu^+$ and its complexation with acetylene into cuprous acetylene [20,23]. However, when $Cu_2O$ is used as the precursor, there is a competition between the reduction of $Cu_2O$ to metallic Cu and the formation of cuprous acetylene (Figure 13). The apparent activation energy for the reduction of $Cu_2O$ is higher than that of CuO, which makes the reduction of $Cu_2O$ to metallic Cu by formaldehyde more difficult, but it is still inevitable [54]. The reduction of $Cu_2O$ to metallic Cu is inhibited by adding $Cl^-$ to $Cu_2O$. $Cu^+$ exists stably under the reaction conditions, and efficiently complexes with acetylene to form cuprous acetylene, which is also stable under the reaction conditions.

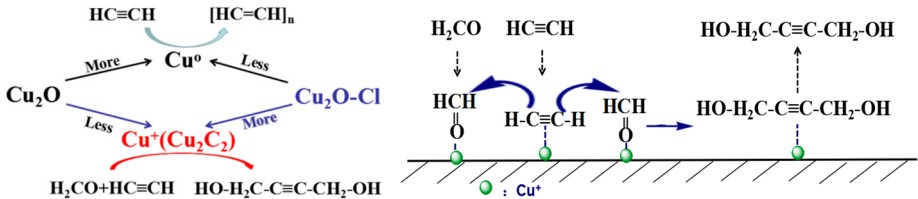

**Figure 13.** A scheme of $Cu_2O$ phase transformation and a plausible reaction mechanism.

Thereby, we believe that $HC{\equiv}C^-$ became adsorbed on the $Cu^+$ site and attacked the electropositive carbon in the formaldehyde adsorbed on the adjacent $Cu^+$. As a result, a new C–C bond was formed, and the final product, BD, was obtained (Figure 13). Considering the low solubility of acetylene in formaldehyde aqueous solution, formaldehyde was always present in excess on the catalyst surface. It is very rare for acetylene to react with only one molecule of formaldehyde to obtain propynol. This can also explain why propynol was not detected in this work.

## 5. Conclusions

$Cl^-$-modified $Cu_2O$ microparticles with different Cl content were successfully synthesized with copper nitrate as a Cu precursor and NaCl as a $Cl^-$ source. $Cl^-$ in the $Cl^-$-modified $Cu_2O$ microparticles decreased the amount of adsorbed oxygen and enhanced the Cu–O bond strength. As a result, $Cu_2O$ was difficult to reduce, and $Cu^+$ persisted under the reductive reaction conditions. $Cu_2O$-Cl(2) showed the highest catalytic activity and stability in the ethynylation of formaldehyde. For $Cu_2O$-$NO_3$, a large number of $Cu^+$ ions were converted into metallic Cu because of the high reducibility of $Cu_2O$. The production of metallic Cu led to the formation of polyacetylene and reduction of active $Cu^+$ centers, causing the catalyst to exhibit low catalytic performance. This investigation may guide the design and development of novel ethynylation catalysts for 1,4-butynediol synthesis and the understanding of their catalytic roles.

**Author Contributions:** The idea was conceived by H.L. and J.G. J.G. performed the experiments and drafted the paper under the supervision of H.L. M.D., G.Y., Z.L. and Z.W. helped to collect and analyze some characterization data. The manuscript was revised through the comments of all authors. All authors have approved the final version of the manuscript.

**Funding:** This research was funded by the National Natural Science Foundation of China (No. 21503124, U1710221) and the International Scientific and Technological Cooperation Project of Shanxi Province, China (No. 201703D421034).

**Acknowledgments:** The XAFS beam time was granted by 1W1B end-station of the Beijing Synchrotron Radiation Facility, Institute of High Energy Physics, Chinese Academy of Sciences. The staff members of 1W1B are acknowledged for their support in the measurements and data analysis.

**Conflicts of Interest:** The authors declare no conflict of interest.

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
