# Peer review of "High-Performance Chlorine-Doped Cu2O Catalysts for the Ethynylation of Formaldehyde"

_processes, doi:10.3390/pr7040198_

Round 1

Reviewer 1 Report

Authors Cl-modified Cu2O microcrystals were prepared using copper nitrate as a copper source and NaCl as a Cl source. The effects of Cl on the structure, reducibility and catalytic  performance of Cu2O in the ethynylation of formaldehyde were studied Perhaps the most interesting is production of metallic Cu led to the formation of polyacetylene and reduction of active Cu+ centers, causing the catalyst to exhibit low catalytic performance. I think the paper is interesting and the idea of the article is interesting and clear. However, some aspects need to be addressed.
I have the following observations: 

1.   In the Abstract section, the most significant results should be provided. Authors should rewrite the abstract with a clear message.

2.     Authors should mention the advantages of the ethynylation of formaldehyde compared to other 1,4-butynediol  production methods in the introduction section.

3.     Why did authors select this type of catalyst with this content of Cu for this process?

4.     I assume there are some hydrocarbons heavier than CH4 in the product stream. Did authors consider these organic compound?

5.     Did authors investigate the repeatability of experiments?

6.     There are some typos and grammatical errors. Authors should spend time to revise the article from grammatical errors

7. The literature survey needs improvement. I permit to suggest you the following references: Energy Conversion and Management,166, 2018, 268-280; Chem. Commun., 2018, 54, 6484-6502; Chemical Engineering Research and Desige, 132, 2018, 843-852.

8. abbreviations should be expanded at first mention. Authors should define all acronyms in the manuscript.

9. Authors should define Fig 1 B in the caption.

Author Response

Response to Reviewer 1 Comments

Point 1: In the Abstract section, the most significant results should be provided. Authors should rewrite the abstract with a clear message.

Response 1: We rewrote the abstract according to the reviewer’s opinion. The changes have been marked in the manuscript.

Point 2: Authors should mention the advantages of the ethynylation of formaldehyde compared to other 1,4-butynediol production methods in the introduction section.

Response 2: As far as we know, ethynylation of formaldehyde is the only way to synthesize 1,4-butynediol in industry. Also, we have not seen any reports on other methods for the synthesis of 1,4-butynediol in the literature. Therefore, we summarized the current state of formaldehyde ethynylation research in the introduction section.

Point 3: Why did authors select this type of catalyst with this content of Cu for this process?

Response 3: As we described in the introduction, Cu+ is the active center for formaldehyde ethynylation. Maintaining the stability of Cu+ and preventing its excessive reduction to inactive metallic Cu in the reaction are the keys to a highly active ethynylation catalyst. To investigate the effect of Cl ions on the stability and ethynylation performance of Cu+, we used pure cuprous oxide to eliminate the interference of supports and other additives, which is conducive to a clearer understanding of the influence of Cl ions.

Point 4: I assume there are some hydrocarbons heavier than CH4 in the product stream. Did authors consider these organic compound?

Response 4: In formaldehyde ethynylation, acetylene is the mobile phase and excessive. We paid more attention to the conversion of formaldehyde and product distribution in the liquid phase. Although the polymer products of acetylene polymerization were observed on the surface of the catalyst, we did not observe heavier hydrocarbon products in the gas phase as proposed by the reviewer. It is presumed that acetylene is strongly adsorbed on the surface of the catalyst and cannot be desorbed after carbon–carbon coupling occurs. This leads to the formation of long carbon-bond polymers.

Point 5: Did authors investigate the repeatability of experiments?

Response 5: For the data listed in this work, we have repeated experiments to confirm the accuracy of the data. The data of each repeated experiment are similar to those of the previous ones. Therefore, the data and conclusions are reliable. 

Point 6: There are some typos and grammatical errors. Authors should spend time to revise the article from grammatical errors

Response 6: Before submitting the manuscript, we had a professional editorial team polish it. According to the suggestion of the reviewer, we revised the manuscript again.

Point 7: The literature survey needs improvement. I permit to suggest you the following references: Energy Conversion and Management,166, 2018, 268-280; Chem. Commun., 2018, 54, 6484-6502; Chemical Engineering Research and Desige, 132, 2018, 843-852.

Response 7: In line with the reviewer’s opinion, we quoted the three references mentioned above and labeled them 51, 52, and 53, respectively. The corresponding order of other references has been adjusted in the revised manuscript.

Point 8: abbreviations should be expanded at first mention. Authors should define all acronyms in the manuscript.

Response 8: According to the reviewer’s opinion, we re-examined the article and revised the following two abbreviations: Line 94 “X-ray diffraction (XRD)”, and Line 95 “X-ray fluorescence spectroscopy (XRF)”.

Point 9: Authors should define Fig 1 B in the caption.

Response 9: According to the reviewer’s opinion, we defined Figure 1B in the caption of Figure 1 and marked the correction in the manuscript.

Reviewer 2 Report

Paper provides a substantial structural and morphological characterization of Cu2O catalysis for the ethynylation of formaldehyde. The experimental data in the paper would be beneficial in catalysis society. However, the manuscript is little confusing to read and more information about experimental procedure is required to avoid confusion.

1.     In the section 2.1. (Preparation of Cu2O catalysts), the authors should provide the source of the raw materials used in the experiment (company name, batch number etc.). The authors also should provide more information about reaction environments (pressure and gas environment)

2.     In the section 2.1. (Preparation of Cu2O catalysts), explanation of the sample preparation of Cu2O catalyst with different Cl- is confusing. Authors should update the section to make it more clear for the readers.

3.      In the section 2.2, the authors wrote the characterization techniques with separate paragraphs for each sentence. They should combine them to make into one paragraph.

4.     Page for line 141 -156. The authors presented diffraction data for Cu2O catalyst with and without Cl- content.  Table 1 clearly shows that lattice spacing is larger when the catalyst does not have any Cl- content. However, in the paragraph (Line 141-156), the authors claimed that Cl- content increases the lattice parameter of the catalyst. This is completely opposite than what they showed in the Table 1. This is very crucial for their claim about the possible reason behind the improvement in catalytic activity of the catalyst with Cl- presence. The authors must clarify this conflict between data and their discussion. I do not suggest this paper to be published until this issue is clarified.

5.      In lines 167-167, the authors need to elaborate their conclusion about the source of Cu and Cl- by using FTIR and Raman data.

6.     In line 251, the authors should explain the experimental details of the H2-TPR characterization.

7.     In line 281, the authors used terminology of “used catalyst”. They need to explain the conditions the catalyst were exposed to. Also, they need to update figure 8 and 9 titles similarly.

8.     In Figure 11, the authors should explain how they calculated the yield of BD.

9.     Discussion section is weak and does not support the suggested mechanisms.  The authors need to be more clear and direct about their suggested mechanism and they should show the experimental evidence behind of their suggested mechanism more directly.  

Author Response

Point 1:  In the section 2.1. (Preparation of Cu2O catalysts), the authors should provide the source of the raw materials used in the experiment (company name, batch number, etc.). The authors also should provide more information about reaction environments (pressure and gas environment)

Response 1: According to the reviewer’s opinion, we added the information of the source of raw materials used in the experiment and marked the correction in the manuscript. The ethynylation reaction was carried out at atmospheric pressure, and this information was added to section 2.1.

Point 2:In the section 2.1. (Preparation of Cu2O catalysts), explanation of the sample preparation of Cu2O catalyst with different Cl- is confusing. Authors should update the section to make it more clear for the readers.

Response 2: According to the reviewer’s opinion, we revised the explanation of sample preparation and marked the correction in the manuscript. 

Point 3: In the section 2.2, the authors wrote the characterization techniques with separate paragraphs for each sentence. They should combine them to make into one paragraph.

Response 3: According to the reviewer’s opinion, we merged the descriptions of characterization techniques into one paragraph and marked the correction in the manuscript.

Point 4: Page for line 141 -156. The authors presented diffraction data for Cu2O catalyst with and without Cl- content.  Table 1 clearly shows that lattice spacing is larger when the catalyst does not have any Cl- content. However, in the paragraph (Line 141-156), the authors claimed that Cl- content increases the lattice parameter of the catalyst. This is completely opposite than what they showed in the Table 1. This is very crucial for their claim about the possible reason behind the improvement in catalytic activity of the catalyst with Cl- presence. The authors must clarify this conflict between data and their discussion. I do not suggest this paper to be published until this issue is clarified.

Response 4: Thank you for your careful review. In the previous format revision, we mistakenly deleted the “d-spacing” column in Table 1. The d-spacing for (111) plane of the Cl-modified Cu2O was higher than that of Cu2O-NO3, which indicates the expansion of the lattice. We added this information into the new manuscript and marked it in red font in Table 1. We are very sorry for the dilemma caused by this.

Point 5: In lines 167-167, the authors need to elaborate their conclusion about the source of Cu and Cl- by using FTIR and Raman data.

Response 5: Only characteristic peaks attributed to Cu2O were observed in the FTIR and Raman data. No bonding with Cl was detected because of its low content and the sensitivity of the two methods.

Point 6: In line 251, the authors should explain the experimental details of the H2-TPR characterization.

Response 6: According to the reviewer’s opinion, we supplemented the experimental details of H2-TPR characterization in Section 2.2 and marked the correction in the manuscript. Also, we added the explanation of the reduction peak attribution as follows: “Since there were no other reducible metal oxide components in the three Cu2O samples, the hydrogen consumption peaks of each sample corresponded to the reduction of Cu+ or a small amount of Cu2+ to metallic Cu” in Section 3.1.5.  

Point 7: In line 281, the authors used terminology of “used catalyst”. They need to explain the conditions the catalyst were exposed to. Also, they need to update figure 8 and 9 titles similarly.

Response 7: According to the reviewer’s opinion, we described the treatment conditions of the used catalyst before characterization in lines 291 and 292. Also, we updated the titles of Figures 8, 9, and 10.

Point 8:  In Figure 11, the authors should explain how they calculated the yield of BD.

Response 8: According to the reviewer’s opinion, we explained the calculation of BD yield in Section 2.3.

Point 9:  Discussion section is weak and does not support the suggested mechanisms. The authors need to be more clear and direct about their suggested mechanism and they should show the experimental evidence behind of their suggested mechanism more directly.

Response 9: In the discussion section, we summarized the information on the characterization of catalysts before and after the reaction around the questions raised in the introduction (whether Cl doping affects the stability and catalytic performance of Cu+ in the ethynylation of formaldehyde). We also focused on the effects of Cl on the structure and reducibility of Cu2O, and the structure–activity relationship in the ethynylation of formaldehyde. The following conclusions were drawn: “The reduction of Cu2O to metallic Cu is inhibited by adding Cl to Cu2O. Cu+ exists stably under the reaction conditions and efficiently complexes with acetylene to form cuprous acetylene, which is also stable under the reaction conditions. As a result, the activity and stability of Cu2O were significantly improved.” This investigation may guide the design and development of novel ethynylation catalysts for 1,4-butynediol synthesis and the understanding of their catalytic roles.

We proposed a possible reaction mechanism based on the literature and the understanding of the regularity of copper species transformation in the reaction process. The mechanism explains the effect of Cl on the ethynylation performance of the catalyst. As the reviewer said, direct evidence for this mechanism in this work is weak. In fact, the understanding of the reaction mechanism is insufficient in the current reports on formaldehyde ethynylation, and no direct evidence of the reaction mechanism has been given. This is due to the complexity of formaldehyde ethynylation reaction system (in situ conversion of active components, the formation of acetylene in an amorphous and highly unstable state, the symmetry of acetylene molecules, the interference of solvent water, etc.). We are preparing a follow-up study to understand the micro-chemical mechanism of formaldehyde ethynylation further. In that study, we are trying to use a variety of in situ characterization techniques (IR, SERS, XPS, XRD, TEM, etc.) and theoretical calculations to explore the formation process of the catalyst microstructures as well as the transformation morphology of reaction molecules and intermediate species. The work will be published later.

Round 2

Reviewer 1 Report

Since the authors have addressed all of my questions and concerns in the revised manuscript, please be informed that the article can be published as it is.

Reviewer 2 Report

The authors updated the paper with the previous suggestions. I think paper contains important experimental results for the community.